

# The rhizosphere and root selections intensify fungi-bacteria interaction in abiotic stress-resistant plants

Feng Huang[1], Mengying Lei[2] and Wen Li[3]

[1] Plant Protection Research Institute, Guangdong Academy of Agricultural Sciences, Key Laboratory of Green Prevention and Control on Fruits and Vegetables in South China Ministry of Agriculture and Rural Affairs, Guangdong Provincial Key Laboratory of High Technology for Plant Protection, Guangzhou, Guangdong, China
[2] Guangdong Eco-Engineering Polytechnic, Guangzhou, Guangdong, China
[3] Key Laboratory of Plant Development and City College of Vocational Technology·Utilization of Ningbo, Ningbo, Zhejiang, China

Corresponding author
Feng Huang, rm12407@126.com

## ABSTRACT

The microbial communities, inhabiting around and in plant roots, are largely influenced by the compartment effect, and in turn, promote the growth and stress resistance of the plant. However, how soil microbes are selected to the rhizosphere, and further into the roots is still not well understood. Here, we profiled the fungal, bacterial communities and their interactions in the bulk soils, rhizosphere soils and roots of eleven stress-resistant plant species after six months of growth. The results showed that the root selection (from the rhizosphere soils to the roots) was stronger than the rhizosphere selection (from the bulk soils to the rhizosphere soils) in: (1) filtering stricter on the fungal (28.5% to 40.1%) and bacterial (48.9% to 68.1%) amplicon sequence variants (ASVs), (2) depleting more shared fungal (290 to 56) and bacterial (691 to 2) ASVs measured by relative abundance, and (3) increasing the significant fungi-bacteria crosskingdom correlations (142 to 110). In addition, the root selection, but not the rhizosphere selection, significantly increased the fungi to bacteria ratios (f:b) of the observed species and shannon diversity index, indicating unbalanced effects to the fungal and bacteria communities exerted by the root selection. Based on the results of network analysis, the unbalanced root selection effects were associated with increased numbers of negative interaction (140 to 99) and crosskingdom interaction (123 to 92), suggesting the root selection intensifies the negative fungi-bacteria interactions in the roots. Our findings provide insights into the complexity of crosskingdom interactions and improve the understanding of microbiome assembly in the rhizosphere and roots.

# INTRODUCTION

The plant underground area, including plant roots and their surrounding soils, is abundantly colonized by different microbes, especially by abundant fungal and bacteria species (*Mitter, De Freitas & Germida, 2017*; *Zhalnina et al., 2018*). According to the

physical proximity to the plant surface and the level of host influence on microbial communities, this area is separated to two different compartments: the rhizosphere and the root endosphere (*Fitzpatrick et al., 2018*; *Yurgel et al., 2018*). It is supposed that the plant deploys two layers of selection to recruit and assemble the microbial communities inhabiting in these two compartments: the rhizosphere communities are formed with microbes mainly from the bulk soils, while the root endosphere communities are formed with microbes mainly from the rhizosphere (*Reinhold-Hurek et al., 2015*). Both of the two layers of selection are affected by plant genetics, and in turn, the assembled microbial communities can deliver essential ecosystem services, such as nutrient cycling, soil structuring, and stress alleviation, back to the plant (*Saleem et al., 2018*).

The plant rhizosphere selection is mainly achieved by root exudates, such as many small signaling molecules (*Hassan & Mathesius, 2012*; *Huang et al., 2014*), polymers (*Beauregard et al., 2013*), antimicrobials (*Huang et al., 2014*), and plant hormones (*Lebeis et al., 2015*). In comparison, the root selection is mainly achieved by plant epidermis and plant immunity system (*Reinhold-Hurek et al., 2015*). Both of the selections can alter plant-microbe and microbe-microbe interactions (*Dundore-Arias et al., 2023*; *Zhang et al., 2024*), thus cause the microbial taxal and functional variations to improve plant health and stress resistance (*Yeoh et al., 2017*; *Cordero, de Freitas & Germida, 2020*).

Different microbial interaction types can have different biological implications to their host plant (*Chepsergon & Moleleki, 2023*; *Liu et al., 2023a*). For example, the positive interactions, generally recognized as microbial cooperation, of different fungal species were capable to improve metabolic efficiency in the root endosphere and rhizosphere soils of the plants *Miscanthus sinensis* and *M. floridulus* (*Ji et al., 2023*). The negative interactions, generally recognized as microbial competition, of bacteria in competition for iron are efficient in suppression of potential soil pathogens in the rhizosphere of various crops (*Gu et al., 2020*). In response to environmental stresses and biotic pathogens, the frequency of microbial interactions generally increases, and confers beneficial effects to improve plant resistance (*Ge et al., 2023*; *Zhou et al., 2023*). Many of these microbial interactions are predictable, and can be characterized and synthesized to form brief communities for ecological and agricultural applications (*Wu et al., 2023*). However, the microbial assembly and interactions of most plants, especially those plants with specific ecological and agricultural attributes, are still not completely studied.

In this study, eleven plant species were selected. All these plant species are native plants in Hainan, China, which can adapt to the harsh environment of soil high salinity and alkalinity, high temperature, strong light, and frequent droughts (*Tong et al., 2013*; *Liu et al., 2015*; *Ren et al., 2017*; *Li et al., 2018*). Furthermore, several of these plant species have been experimentally proved with strong abilities in improving soil water and nutritional conditions, and also in resistance to abiotic stresses (*Wang et al., 2019*; *Zhang et al., 2019*; *Li et al., 2021*). Based on these abilities, these plant species have been recommended and used as pioneer plants in restoration of destroyed and degraded ecosystems (*Tong et al., 2013*; *Liu et al., 2015*; *Ren et al., 2017*). Our study focused on the underground fungal and bacterial communities associated with these plant species, and aimed to test whether the

soil fungal, bacteria communities, and their crosskingdom interactions respond differently to the rhizosphere and root selections.

## MATERIALS AND METHODS

### Plant growth and sampling

Eleven plant species (Table S1), including *Casuarina equisetifolia*, *Calophyllum inophyllum*, *Fagraea ceilanica*, *Guettarda speciosa*, *Heritiera littoralis*, *Hernandia sonora*, *Melaleuca bracteata*, *Pongamia pinnata*, *Portulaca pilosa*, *Ruellia brittoniana*, and *Scaevola sericea*, natively grow and thrive under the environment of high soil salinity and alkalinity, high temperature, strong light, and frequent drought in Hainan, China (*Tong et al., 2013*; *Liu et al., 2015*; *Ren et al., 2017*; *Li et al., 2018*). Their seeds were collected in Hainan, and sown in pots (24 × 26 cm, diameter × height) filled with field soils near a greenhouse of Wenchang Lingye Gardening Limited Company, Hainan, China. For each plant species, 24 seeds were selected to germinate, and four seeds were sown in a pot. After germination, only one healthy seedling was left in the pot, and kept growing in the same greenhouse from February to August, 2017. All the plants were harvested on August 14, 2017. The soils collected before planting were designated as the bulk soils. The plant roots were dug out from the pot soils, and then shaken to remove loosely attached soils. Only the soils, closely adhered to the roots, were mechanically brushed and kept as the rhizosphere soils. The fine roots, with diameter smaller than 2 mm, were washed in 1 × TE buffer (added with 0.1% Triton X-100) for 30 s, 75% ethanol for 15 s, 2% bleach for 15 s, and rinsed three times with sterile water (*Agler et al., 2016*). All samples were put into sterile plastic bags and stored under −80 °C.

### DNA extraction and amplicon sequencing

Microbial genomic DNA was extracted from 250 mg soils or 100 mg ground roots using a Powersoil® Kit (MoBio Laboratories Inc., Carlsbad, CA, USA). Briefly, the sample was lysed *via* chemical and mechanical homogenization using lysis buffer from the kit and a standard bench-top vortex. The crude lysate was collected and cleaned up to remove PCR inhibitors. Then, the purified lysate was mixed with an equal volume of DNA binding solution, passed and washed twice through a silica spin filter membrane. The silica-bound DNA was eluted using a 10 mM Tris elution buffer and kept for downstream applications. The partial nucleotide sequences of fungal nuclear ribosomal internal transcribed spacer (ITS rDNA) and bacterial 16S rRNA were amplified by polymerase chain reaction (PCR) with fungal primer set ITS5-1737F (5′-GGAAGTAAAAGTCGTAACAAGG-3′)/ITS2-2043R (5′-GCTGCGTTCTTCATCGATGC-3′) and bacteria primer set 515F (5′-GTGCCAGCMGCCGCGGTAA-3′)/806R (5′-GGACTACHVGGGTWTCTAAT-3′), respectively (*Caporaso et al., 2011*). PCR reaction was carried out using Phusion High-Fidelity PCR Master Mix (New England Biolabs Inc., Ipswich, MA, USA). The PCR cycle was carried out with an initial 94 °C for 5 min, followed by 35 cycles of 94 °C for 45 s, 56 °C for 30 s, 72 °C for 30 s, and a final extension at 72 °C for 10 min. For each sample, the PCR was conducted in triplicate and then pooled to make one PCR mixture. The harvested DNA was verified on 2% agarose electrophoresis gel and purified with Qiagen Gel

Extraction Kit (Qiagen, Germany). Sequencing libraries were generated using TruSeq® DNA PCR-Free Library Preparation Kit (Illumina, San Diego, CA, USA), and assessed on the Qubit@ 2.0 Fluorometer (Thermo Fisher Scientific, Waltham, MA, USA) and Agilent Bioanalyzer 2100 system (Agilent Technologies, CA, USA). The amplicon sequencing was carried out on the Illumina HiSeq2500 platform (Illumina, San Diego, CA, USA) by Novogene Co., LTD (Novogene, Beijing, China).

## Bioinformatics analysis

After sequencing, the raw reads were filtered by Trimmomatic version 0.39 (*Bolger, Lohse & Usadel, 2014*) and cutadapt v1.9.1 (*Martin, 2011*) to remove low-quality reads (with more than 10% unknown nucleotides, and less than 80% Q-value >20 bases), short sequences (<200 bp), adapters, primers, and poly bases. The left high-quality reads were assembled by FLASH (v1.2.7, http://ccb.jhu.edu/software/FLASH/) with a minimal overlap of 16 bp and mismatch of 5 bp. De-noise and removal of chimeric sequences were processed by dada2 (*Callahan et al., 2016*) in QIIME2 2020.6 (*Bolyen et al., 2019*). The amplicon sequence variants (ASVs) were clustered by dada2, and annotated by bayesian classifier using UNITE (https://unite.ut.ee/) and SILVA138 (http://www.arb-silva.de/) as fungal and bacterial reference database, respectively.

## Network analysis

The microbial interaction network was constructed using SParse InversE Covariance Estimation for Ecological Association Inference (SPIEC-EASI) R package v1.0.7 (*Kurtz et al., 2015*). For microbial network inference, ASVs with an average relative abundance lower than 0.02% were filtered out and only top fungal and bacterial ASVs were used. All networks were constructed following the same workflow. The specific construction workflow could be found in an R document (https://www.rdocumentation.org/packages/SpiecEasi/versions/1.0.7), the parameters were set as default. The network layout was shown, and also the network descriptive parameters was calculated, in R package igraph v1.3.2 (https://igraph.org/) based on the Fruchterman-Reingold algorithm.

## Statistical analysis

The ASVs table for all samples was generated in QIIME2 2020.6 (*Bolyen et al., 2019*), and then normalized to the sample with the least sequence number (39,262 for ITS and 37,161 for 16S rRNA sequences; Fig. S1), the alpha diversity indices (Observed species, Shannon, Simpson, ACE, Chao1, Good's coverage, and Phylogenetic diversity) of each sample and the relative abundances of each taxon were subsequently calculated in the same software. All statistical analyses were conducted in the R software version 4.0.0 (*R Core Team, 2022*). The estimation of fold change of fungal and bacteria ASVs was performed in DESeq2 (*Love, Huber & Anders, 2014*). The correlation analysis was carried out in the package Hmisc (https://hbiostat.org/R/Hmisc/). The data presented by heatmap were carried out and plotted in the package pheatmap (*Kolde, 2019*). The ANOVA followed by Tukey *post-hoc* pairwise test was used for comparisons among the bulk soils, rhizosphere soils, and

roots. The *P* values were adjusted using false discovery rate (fdr) method, *P* values smaller than 0.05, except for the correlation analysis, were accepted with significance.

## RESULTS

### The rhizosphere and root selections on fungal and bacterial communities

A total of 97 samples, including 17 samples of bulk soils, 42 of rhizosphere soils, and 38 of plant roots, were successfully sequenced with both qualified ITS and 16S rRNA sequences (Table S1; Figs. S1A and S1B, respectively). Accordingly, 7,142,267 ITS sequences (mean 73,632 ± standard deviation 12,779 per sample) and 6,937,220 16S rRNA sequences (71,518 ± 11,483) were harvested, respectively. After normalization (at 39,262 for ITS and 37,161 for 16S rRNA sequences) and deletion of singletons, the sequences were clustered into 15,626 (989 ± 448) fungal and 23,423 (999 ± 578) bacteria ASVs, respectively.

By compartment, 2,978 (19.1% of all fungal ASVs, Fig. 1A) fungal and 9,762 (41.7% of all bacteria ASVs, Fig. 1B) bacteria ASVs were found in the bulk soils, rhizosphere soils, and roots. In addition, 6,268 (2,978 + 3,290, 40.1%) fungal and 15,942 (9,762 + 6,180, 68.1%) bacteria ASVs were found in bulk soils and rhizosphere soils, only 1, 110 (1,033 + 77, 7.1%) fungal and 1,449 (1,291 + 158, 6.2%) bacteria ASVs found in bulk soils were not found in rhizosphere soils; 4,461 (2,978 + 1,483, 28.5%) fungal and 11,445 (9,762 + 1,683, 48.9%) bacteria ASVs were found in rhizosphere soils and roots, 9,212 (5,922 + 3,290, 59%) fungal and 10,178 (3,998 + 6,180, 43.4%) bacteria ASVs found in rhizosphere soils were not found in roots.

The mean numbers of both the fungal and bacteria ASVs decreased significantly from bulk soils to rhizosphere soils (1,411 ± 271 to 1,220 ± 286, $P$ = 0.03 for fungi; 1,617 ± 323 to 1,277 ± 367, $P$ < 0.01 for bacteria), and further from rhizosphere soils to roots (to 546 ± 234, $P$ < 0.01; to 416 ± 249, $P$ < 0.01). The fungi to bacteria ratios (f:b) of both the observed species and shannon diversity index were not significantly different between bulk soils and rhizosphere soils, but significantly improved from rhizosphere soils to roots (1.0 ± 0.4 to 1.6 ± 0.8, $P$ < 0.01 for f:b of the observed species, Fig. 1C; 0.6 ± 0.1 to 0.8 ± 0.3, $P$ < 0.01 for f:b of the shannon diversity index, Fig. 1D).

### The rhizosphere and root selections on fungal and bacterial ASVs

After the dropout of low abundant ASVs (the mean relative abundance <0.01%), 2,249 fungal and 4,523 bacteria ASVs were used for the comparisons between bulk soils and rhizosphere soils, and between rhizosphere soils and roots. Following the criteria (|$\text{Log}_2\text{FoldChange}$| ≥ 2 & adjusted $P$ value < 0.01), 56 fungal and two bacteria ASVs were depleted, while 17 fungal and nine bacteria ASVs were enriched in rhizosphere soils in comparison to bulk soils (Figs. 2A and 2B). In addition, 290 fungal and 691 bacteria ASVs were depleted, while 75 fungal and 12 bacteria ASVs were enriched in roots in comparison to rhizosphere soils (Figs. 2C and 2D). Not only the relative abundance of the fungal and bacteria ASVs, the correlation between the fungal and bacteria ASVs was also affected by rhizosphere and root selections. Following the criterion (adjusted $P$ value < 0.01), the number of significant correlations between the top 50 fungal and the top 50 bacteria ASVs

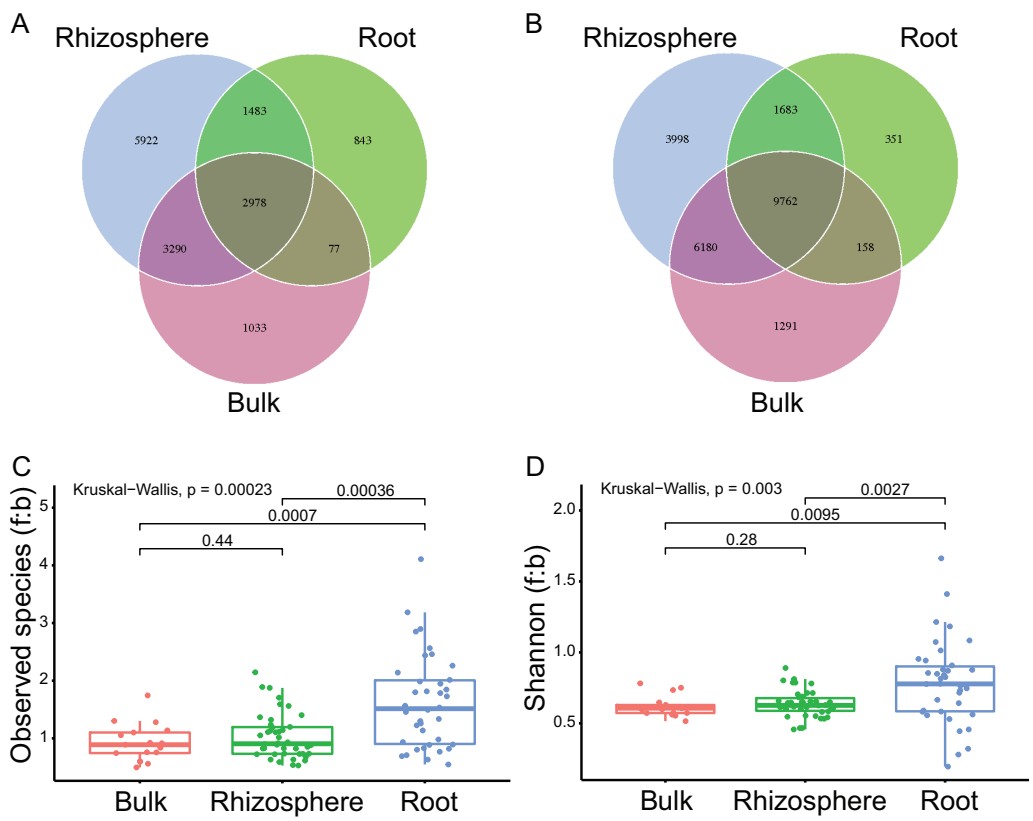

**Figure 1** **The compartment effects on the fungal and bacteria communities.** (A and B) The shared and unique fungal (A) and bacteria (B) ASVs among the bulk soils, the rhizosphere soils, and the roots. (C and D) The fungi to bacteria ratios of observed species (C) and shannon index (D) among the bulk soils, the rhizosphere soils, and the roots. The *P* values between compartments are provided on the line.

increased from 99 in bulk soils to 110 in rhizosphere soils, and further increased to 142 in roots (Fig. 3).

## The rhizosphere and root selections on microbial interaction networks

The interaction networks were constructed for the communities of fungi (*n* = 92, Fig. 4A), bacteria (*n* = 83, Fig. 4B), and combined fungi and bacteria (*n* = 175, Fig. 4C) in bulk soils (left), rhizosphere soils (middle), and roots (right), respectively. For all the three communities, the complexity of the networks increased after rhizosphere and root selections (Fig. 4). In specific, the density of the fungal networks increased from 0.017 in bulk soils to 0.028 in rhizosphere soils, and further increased to 0.035 in roots; the density of the bacteria networks increased from 0.021 to 0.023, and further to 0.025; the density of the combined network increased from 0.021 to 0.026, and further to 0.028 (Fig. 5A). Similarly, the number of edges of the fungal networks increased from 72 in bulk soils to 119 in rhizosphere soils, and further increased to 145 in roots (Fig. 5B); the number of edges of the bacteria networks increased from 71 to 79, and further to 84 (Fig. 5C); the number of edges of the combined networks increased from 319 to 396, and further to 430 (Fig. 5D).

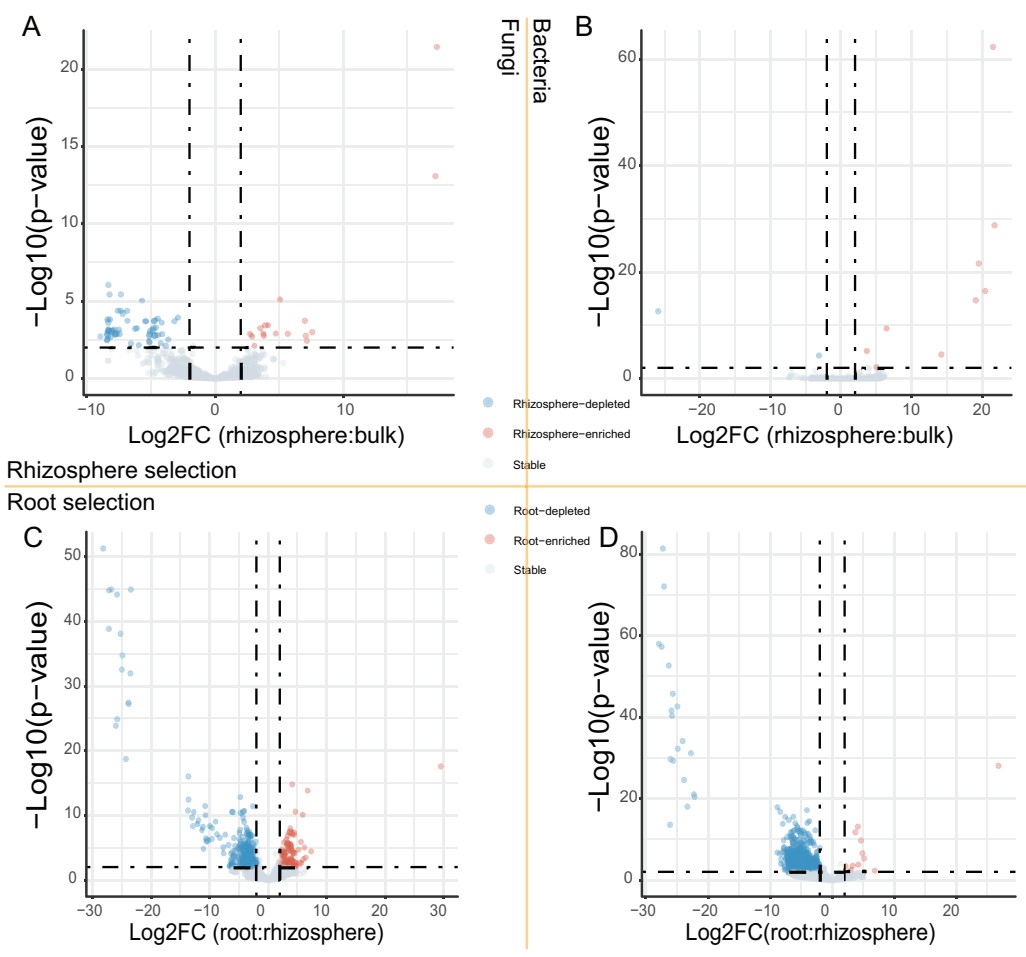

**Figure 2 The volcano plots of fungal and bacteria ASVs differentially presented in different compartments.** (A and B) The fungal (A) and bacteria (B) ASVs varied (enriched, depleted, or stable) in the rhizosphere soils in comparison to the bulk soils. (C and D) The fungal (C) and bacteria (D) ASVs varied in the roots in comparison to the rhizosphere soils. The criteria for significant variation between compartments were set with |Log$_2$FoldChange| ≥ 2 & adjusted $P$ value < 0.01.

As the edge was represented by microbe-microbe interaction, all the edges were divided into either positive (orange) or negative (light blue) microbial interactions. All the microbial networks were dominated by positive interactions (Fig. 4), the mean ratio of positive interactions was 74.2% ± 9%, and the mean ratio of negative interactions was 25.8% ± 9% (Figs. 5B–5D). After the rhizosphere and root selections, both the number of positive (from 61 in the bulk soils to 102 in the rhizosphere soils, and further to 112 in the roots) and negative (from 11 to 17, and further to 33) interactions increased in the fungal networks (Fig. 5B). However, only the numbers of negative interactions increased consistently in the bacteria (from 16 to 26, and further to 36, Fig. 5C) and the combined (from 76 to 99, and further to 140, Fig. 5D) networks. On the contrary, the numbers of positive interaction varied differently in the bacteria (from 55 to 53, and further to 48, Fig. 5C) and the combined (from 243 to 297, and further to 290, Fig. 5D) networks after the rhizosphere and root selections. At last, the variation of the positive and negative

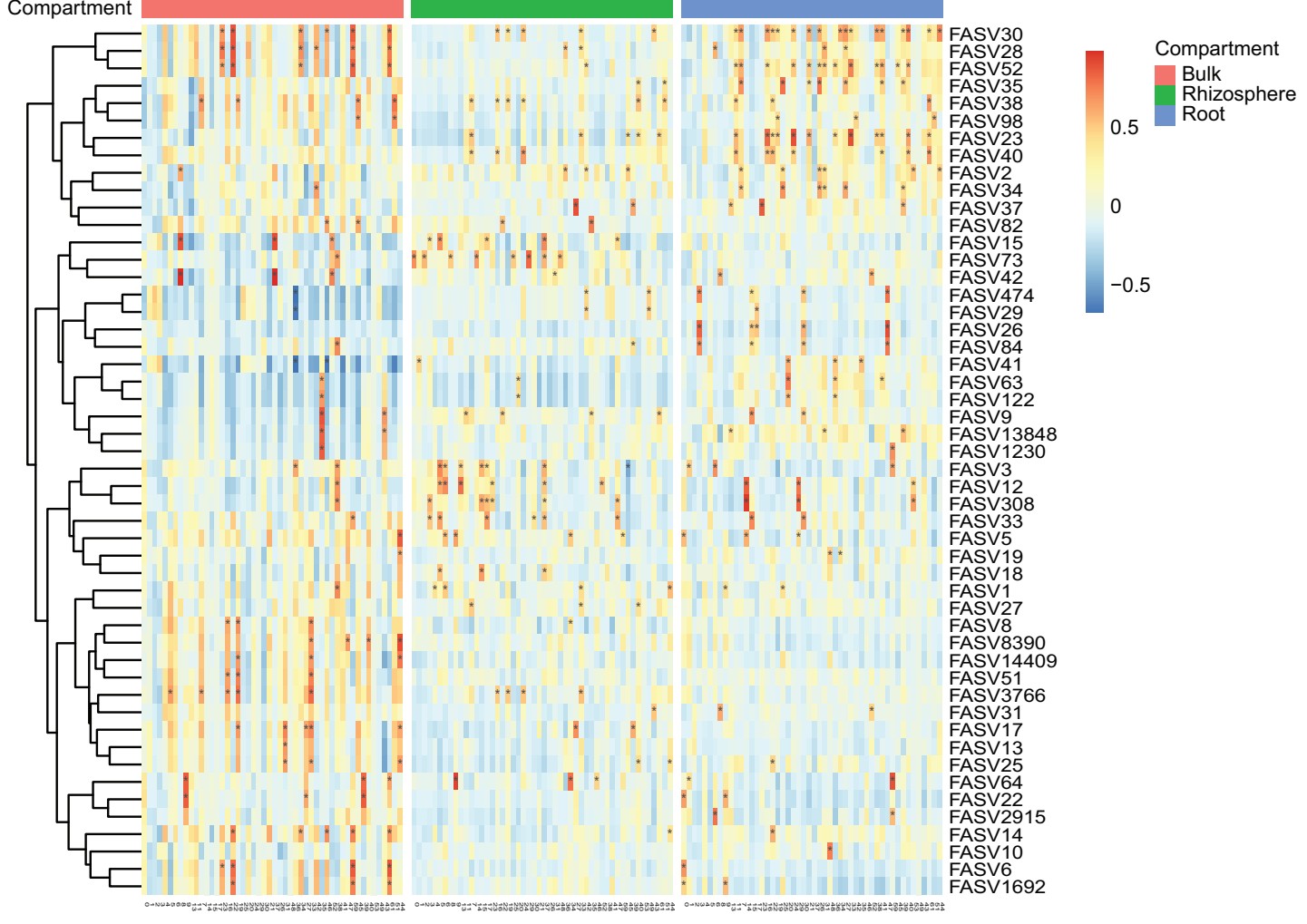

**Figure 3** **The heatmap plots of the correlation coefficients between the top 50 fungal (in row) and bacteria (in column) ASVs in different compartments.** The number of correlation coefficients is represented by the gradient ramp, adjusted *P* value < 0.01 is represented by an asterisk (*).

interactions (Fig. 5D) was similar to the variation of the intra-kingdom and inter-kingdom interactions (Fig. 5E) in the combined communities. After the rhizosphere and root selections, the number of intra-kingdom interaction increased from 231 to 304, and further to 307; the number of inter-kingdom fungi-bacteria interaction increased from 88 to 92, and further to 123.

## DISCUSSION

Hainan Island, with 33, 900 km$^2$ of land area, is the second largest island in China. It locates in the zone of tropical monsoon climate, with an annual average temperature of 23.2–27.1 °C, annual total precipitation of 1,009–2,367.7 mm (nearly 70% in the wet season), and annual sunshine duration of 1,776–2,783 h (*Chen et al., 2022*). For the selected plant species in this study, their native growing soils mainly consisted of phospho-calcic soils and coastal saline soils, which are usually saline, alkaline, and poor in

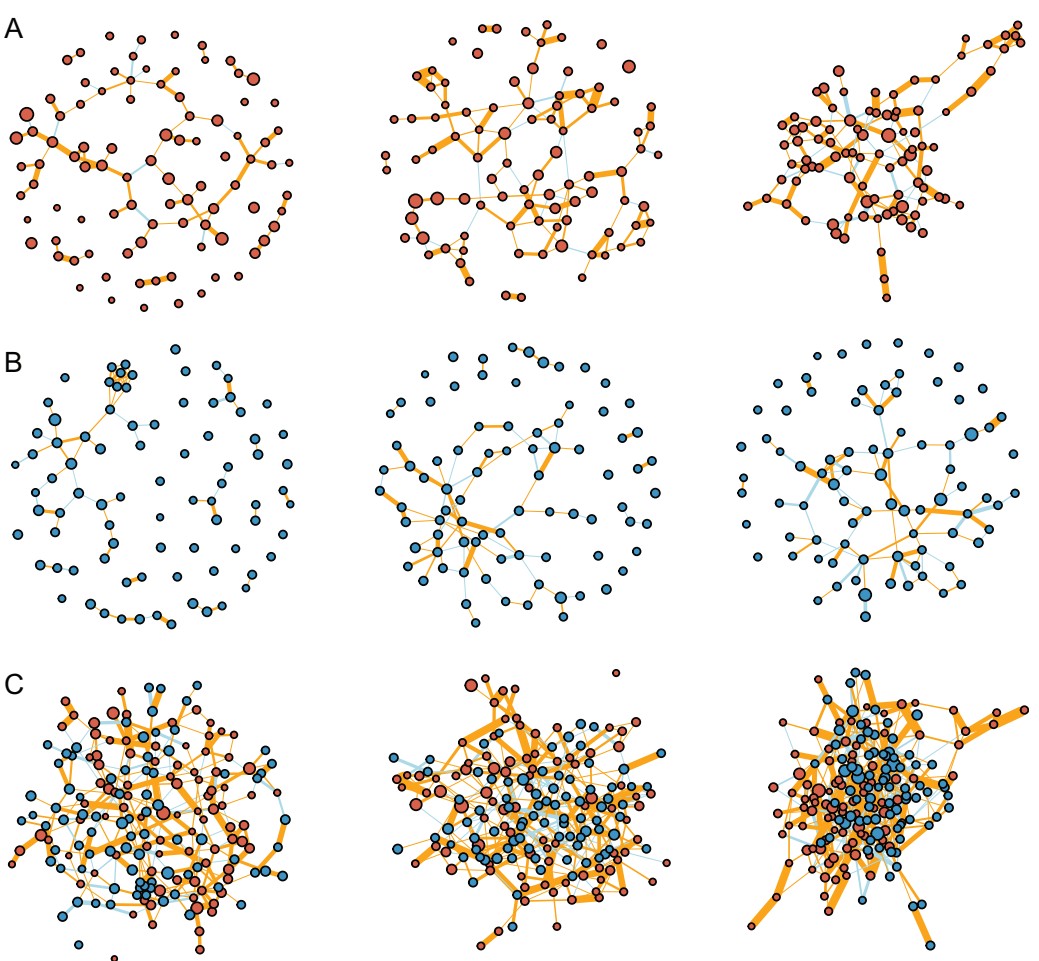

**Figure 4 The inteaction networks of major ASVs constructed for the communities of fungi (A), bacteria (B), and combined fungi and bacteria (C) in the bulk soils (left), rhizosphere soils (middle), and roots (right), respectively.**

content of organic carbons (*Zhou, Zhou & Wang, 2003*; *Li et al., 2018*). Based on their great adaptation to barren lands, these plant species have attracted more and more attentions in the past several years in China (*Li et al., 2018*). For example, *Wang et al. (2019)* demonstrated that the growth of one or several of these plants improved soil quality and vegetation recovery rate by significant increasing of soil contents of water, microbial biomass carbon and nitrogen. Most of all, the plants of *Calophyllum inophyllum* and *Guettarda speciosa* showed strong abilities of stress resistance (*Wang et al., 2019*; *Zhang et al., 2019*; *Li et al., 2021*). However, their root microbiomes, supposed to offer many beneficial traits to plant growth and resistance, have received little attention.

Our study offered a snapshot of the assembly of fungal and bacterial communities after the abiotic stress-resistant plants growth for about six months. It proved that both fungal and bacterial communities respond more drastically to the root selection than the rhizosphere selection (*Reinhold-Hurek et al., 2015*). The shared fungal and bacterial ASVs were more frequently depleted by the root selection than by the rhizosphere selection (Fig. 2). With abundant root exudates in the rhizosphere area, many different microbes can

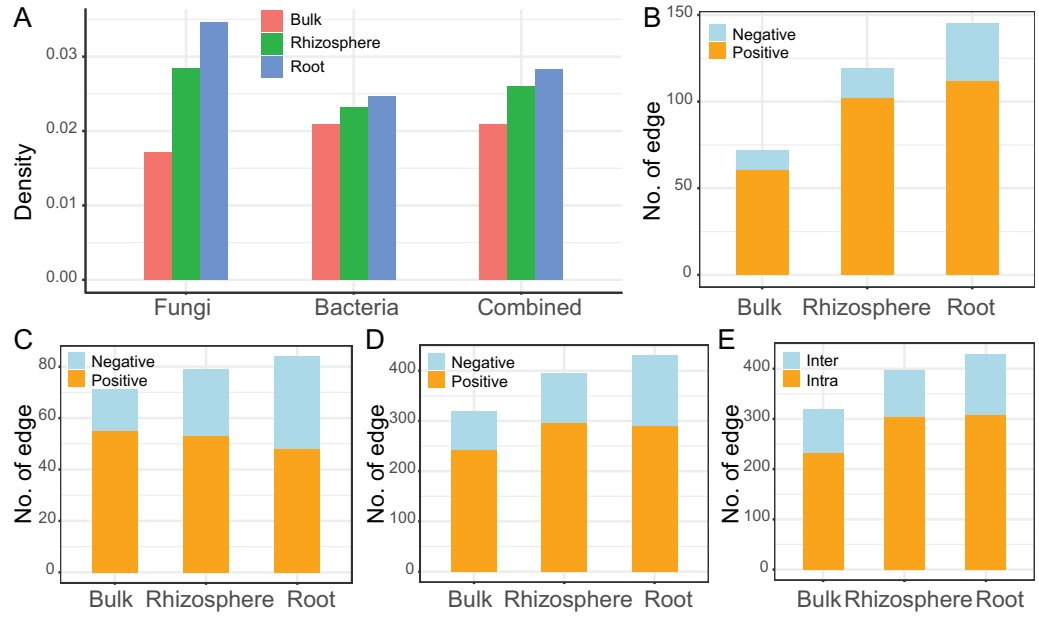

**Figure 5 The descriptive parameters of the microbial interaction networks.** (A) The density value of the fungal, bacteria, and the combined communities of different compartments. (B–D) The number of positive and negative edges in the networks of the fungal (B), bacteria (C), and the combined communities (D). (E) The number of interkingdom and intrakingdom edges in the network of the combined community.

co-exist and work synergistically to transform unusable substrates to usable substrates and reduce toxic substances (*McLaughlin et al., 2023*; *Wu et al., 2023*). The usable substrates can further support the growth of many microbial saprophytes, which results in the relatively more abundant fungal and bacterial communities in the rhizosphere (*Hassan & Mathesius, 2012*; *Trivedi et al., 2020*). In contrast, the root endosphere is protected by a physical barrier (root epidermis) and the plant defense system, which results in the relatively less abundant fungal and bacterial communities in the root (*Hassan & Mathesius, 2012*; *Trivedi et al., 2020*). In addition, the priority effect, namely, early colonized microbes affect the colonization of microbes that arrive later, may also affect the abundances of fungal and bacterial communities in the root (*Debray et al., 2022*).

Furthermore, we found that the fungi to bacteria ratios (f:b) of both the observed species and shannon diversity index were significantly increased by root selection other than by rhizosphere selection (Figs. 1C and 1D). These results suggest that: (1) root selection may have weaker effects on fungal communities and/or stronger effects on bacterial communities; (2) root selection may facilitate the growth of fungal endophytes which can inhibit the growth of some bacteria. Many fungal endophytes can establish a mutualistic relationship with plant roots, in which the fungi can improve the growth and stress resistance of their host plant through mechanisms such as phosphorus transmission, stimulation of proline, glycine betaine production, increase of plant indoleacetic acid concentration *etc*. (*Liu et al., 2023b*; *Miranda et al., 2023*; *Toppo et al., 2023*). In a recent study, *Ma et al. (2023)* showed that the mutualistic growth of a fungal endophyte (*Phomopsis liquidambaris*) in the plant (*Arachis hypogaea* L.) roots not only promoted

host growth and disease control, but also reshaped the core root bacterial taxa. Based on these, one explanation of the unbalanced effects of root selection on fungal and bacterial communities may be that the roots of our stress-resistant plants associate closely with their endosphere fungal communities, and then, some fungal endophytes reshape the bacterial communities through crosskingdom interactions.

Based on the correlation and network analysis, we found that the microbial total interaction increased in number and strength in response to the rhizosphere and root selections (Figs. 3 and 4). Firstly, we found that all the microbial communities of bulk soils, rhizosphere soils, and roots were dominated by positive interactions. This may be explained by that the microbial communities rely on cooperative relationships to improve metabolic efficiency and promote soil and root colonization (*Chepsergon & Moleleki, 2023*; *Ge et al., 2023*; *Zhang et al., 2024*), such as the microbial interaction forms of crossfeeding, syntrophy, physical complex formation, *etc.* (*Wu et al., 2023*). In addition, some of the positive interactions may also be caused by spatial coincidence, rather than actual ecological interactions (*Liu et al., 2023a*). Secondly, we found that the increase of negative interactions was consistent across the fungal, bacterial, and the combined microbial communities in response to both of the two selections. Negative interactions, may be representative of microbial competition, antagonism, or predation, are important for the ecological stability of a microbial community (*Ji et al., 2023*; *Liu et al., 2023a*). In bulk and rhizosphere soils, the maintenance of community positivity (more positive interactions) is beneficial for the microbial community to transform transient root exudates and scavenge soil pathogens and hazardous substances (*Ji et al., 2023*; *Li et al., 2023*; *Zhou et al., 2023*). While in plant roots, the longer ecological stability, by increased negative interactions, may be important for the establishment of mutualistic plant-microbe relationships (*Dundore-Arias et al., 2023*; *Ji et al., 2023*). Lastly, we found that the increased number of negative interactions of the combined microbial community correlated to the increased number of inter-kingdom interactions by the root selection (Figs. 5D and 5E). Negative interactions have been proved to primarily occur through inter-kingdom microbial interactions, and are important for plant host survival and maintenance of host-microbiota balance (*Chen et al., 2018*; *Durán et al., 2018*; *Zhang et al., 2024*). The competition between fungal and bacterial species in plant roots may be caused by their common antagonistic relationships and metabolic overlaps (*Pacheco & Vorholt, 2023*; *Zhang et al., 2024*). Based on these, we infer that the increase of both the numbers of negative interactions and inter-kingdom interactions causes the unbalanced influence of the root selection on fungal and bacterial communities. However, more experimental works are still needed to conform these inferences, and to recur the fungi-bacteria interactions for agricultural and ecological applications.

## CONCLUSION

Our study describes the effects of the the rhizosphere and root selections on structuring the fungal and bacterial communities in bulk soils, rhizosphere soils, and roots of a group of stress-resistant plant species. Both of the two selections intensify the crosskingdom fungi-bacteria interaction. Compared to the rhizosphere selection, the root selection is

more intensive and unbalanced by accumulating more crosskingdom and negative interactions.

## ACKNOWLEDGEMENTS

We thank for our colleagues in the collection of soil samples.

### Funding

This work was funded by the financial support from Guangdong Academy of Agricultural Sciences (R2020YJ-YB3005), and the National Natural Science Foundation of China (31600430). Feng Huang received support from the China Scholarship Council (201704910213). The funders had no role in study design, data collection and analysis, decision to publish, or preparation of the manuscript.

### Grant Disclosures

The following grant information was disclosed by the authors:
Guangdong Academy of Agricultural Sciences: R2020YJ-YB3005.
National Natural Science Foundation of China: 31600430.
China Scholarship Council: 201704910213.

### Competing Interests

The authors declare that they have no competing interests.

### Author Contributions

- Feng Huang conceived and designed the experiments, analyzed the data, prepared figures and/or tables, authored or reviewed drafts of the article, and approved the final draft.
- Mengying Lei performed the experiments, authored or reviewed drafts of the article, and approved the final draft.
- Wen Li performed the experiments, analyzed the data, authored or reviewed drafts of the article, and approved the final draft.

### Data Availability

The raw sequences are available at NCBI BioProject: PRJNA797861.

### Supplemental Information

Supplemental information for this article can be found online at http://dx.doi.org/10.7717/peerj.17225#supplemental-information.

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
