# Peer review of "The rhizosphere and root selections intensify fungi-bacteria interaction in abiotic stress-resistant plants"

_PeerJ, doi:10.7717/peerj.17225_

## Round 0.1 · original submission · Major Revisions

Dear authors, please revise the manuscript carefully. Although the manuscript is rejected by one reviewer, I believe that if the authors can address the reviewer's concern, then the revised manuscript can be reconsidered.

·

Basic reporting

The article is well written. However some latest literature can be added to the introduction and discussion part before final acceptance. I congratulate authors for carrying out such study.

Experimental design

Satisfied.

Validity of the findings

Satisfied.

Additional comments

The article is well written. However some latest literature can be added to the introduction and discussion part before final acceptance.

Reviewer 2 ·

Basic reporting

The manuscript “The rhizosphere and root selections intensify fungi-bacteria interaction in abiotic stress-resistant plants” deals with valuable insights into the intricate dynamics of microbial communities in the rhizosphere and their interactions with stress-resistant plant species.
Comment
• Investigating microbial communities in the rhizosphere of stress-resistant plant species contributes to our understanding of plant-microbe interactions under challenging environmental conditions.
• The methodology used to characterize fungal and bacterial communities in the rhizosphere could be further elucidated. Details on sample collection, DNA extraction, sequencing protocols, and data analysis methods would enhance the reproducibility and transparency of the study.
• LN 50-65: Please rewrite with better clarity
• Can you explain how soil microbes are selected and recruited into the rhizosphere and further into the roots of plants, and what factors influence this process?
• The study reports changes in the ratios of shared fungal and bacterial ASVs, as well as alterations in diversity indices in response to root and rhizosphere selections. However, the ecological significance of these changes could be further discussed. How do shifts in ASV ratios and diversity indices relate to ecosystem functioning and plant health?
• What were the main findings regarding the differences in shared fungal and bacterial ASVs, depleted ASVs, and cross-kingdom correlations between rhizosphere soils and roots compared to bulk soils and rhizosphere soils?
• LN 15-188: Please see the grammar and writing of the content.
• The research mentions that the responses of fungal and bacterial communities to root and rhizosphere selections may be explained by changes in negative interactions and cross-kingdom interactions. However, the specific mechanisms underlying these interactions and their consequences for plant-microbe interactions remain speculative. Providing mechanistic insights into how these interactions shape microbial community assembly and function in the rhizosphere would strengthen the study's conclusions.
• The study's findings suggest that microbial communities respond more drastically and unbalancedly to root selection compared to rhizosphere selection, indicating the pivotal role of root exudates and root-microbe interactions in microbiome assembly. I appreciate the author on this aspect.
• LN 223-233: Please rewrite the content with better clarity.
• How were the fungal-to-bacterial ratios and Shannon diversity index altered by root selection compared to rhizosphere selection, and what does this suggest about the responses of microbial communities?
• Understanding the complex interactions between microbial communities and stress-resistant plant species in the rhizosphere has implications for agriculture and ecosystem management. Discussing how these findings could be applied to enhance plant stress resilience, soil health, and ecosystem sustainability would broaden the relevance and impact of the research.

Experimental design

See section 1

Validity of the findings

See section 1

Additional comments

See section 1

Reviewer 3 ·

Basic reporting

Ref: Peer J 95039
Title: The rhizosphere and root selections intensify fungibacteria interaction in abiotic stress-resistant plants

Abstract:
Line no 36. it should be selection, Abstract should be written with key findings, there are many grammatical errors, Author carefully go for entire manuscript.
Keywords: all the keywords should be alphabetical in order following journal format
Introduction
Consider evaluating the manuscript for language and grammar to ensure clarity and readability.
A few of the points are only loosely related to the review's objective.
1. Please carefully read it once more, revise the introduction section, and make an effort to tie it to the main goal of the review. Cite recent papers as well. Although, authors framed the manuscript well but need to improve the description from lines 106-113 to provide more justification of the study.
Materials and methods
Line no. 123.what was the criteria for selection of 11 different plant species, whether author confirmed microbial communities in individual selected plant species. Some tome microbial population and their interaction with plants species varies on different interaction mechanism. Author should be mentioned the number of replicates used during the PCR study, why other molecular method was not used for checking consistency of results accuracy. Author should be described methodology appropriately.
Results
After the rhizosphere and root selections, both the number of positive and negative interaction increased in the fungal networks why ????. while, only negative interaction increased consistently in the bacteria and in combined networks. The datas of fig 1 is not matched in results section, it should be carefully check again. Remove any extraneous stuff and make an effort to connect each of sentence of results.
Discussion
At this stage, it is very difficult to understand discussion part. There are many grammatical errors. Author should rewrite discussion with support of latest references and with conjunction of supporting results. In discussion, to enhance the overall quality of the manuscript, the authors should expand on the literature regarding interaction study of microbes in rhizosphere. Need a comprehensive concluding result section at its end, too.
Conclusion: no conclusion section in manuscript?????
General
The author should check all citation and references in entire manuscript and arrange as per the format of the journal. All the scientific name should be in italics form, name of the journal citation, figure, tables should be uniform.
Decision: I must be recommended rejection.

Experimental design

as per basic reporting

Validity of the findings

Conclusion: conclusion section missing in manuscript?????

Additional comments

as per basic reporting

---

## Round 0.2 · accepted · Accept

The revised manuscript is accepted.

·

Basic reporting

The authors have now revised the MS in accordance to the provided comments. Thus the article may be accepted now in its current form.

Experimental design

Satisfied

Validity of the findings

Satisfied

Additional comments

Not required

Reviewer 3 ·

Basic reporting

Yes, authors incorporated all comments as suggested by all reviewers.

Experimental design

satisfied

Validity of the findings

satisfied

Additional comments

No